# Characterization of the Kenyan Honey Bee (*Apis mellifera*) Gut Microbiota: A First Look at Tropical and Sub-Saharan African Bee Associated Microbiomes

**DOI:** 10.3390/microorganisms8111721

**Published:** 2020-11-03

**Authors:** Yosef Hamba Tola, Jacqueline Wahura Waweru, Gregory D. D. Hurst, Bernard Slippers, Juan C. Paredes

**Affiliations:** 1International Centre of Insect Physiology and Ecology (*icipe*), Nairobi 30772-00100, Kenya; yosef@icipe.org (Y.H.T.); jackiemaingih@gmail.com (J.W.W.); 2Department of Biochemistry, Genetics and Microbiology, Forestry and Agricultural Biotechnology Institute (FABI), University of Pretoria, Pretoria 0002, South Africa; bernard.slippers@fabi.up.ac.za; 3Institute of Infection, Veterinary and Ecological Sciences, University of Liverpool, Liverpool L69 3BX, UK; G.Hurst@liverpool.ac.uk

**Keywords:** *Apis mellifera*, African honey bee, *Gilliamella*, *Frischella*, gut microbiota, microbiome, symbiont

## Abstract

Gut microbiota plays important roles in many physiological processes of the host including digestion, protection, detoxification, and development of immune responses. The honey bee (*Apis mellifera*) has emerged as model for gut-microbiota host interaction studies due to its gut microbiota being highly conserved and having a simple composition. A key gap in this model is understanding how the microbiome differs regionally, including sampling from the tropics and in particular from Africa. The African region is important from the perspective of the native diversity of the bees, and differences in landscape and bee management. Here, we characterized the honey bee gut microbiota in sub-Saharan Africa using 16S rRNA amplicon sequencing. We confirm the presence of the core gut microbiota members and highlight different compositions of these communities across regions. We found that bees from the coastal regions harbor a higher relative abundance and diversity on core members. Additionally, we showed that *Gilliamella, Snodgrassella,* and *Frischella* dominate in all locations, and that altitude and humidity affect *Gilliamella* abundance. In contrast, we found that *Lactobacillus* was less common compared temperate regions of the world. This study is a first comprehensive characterization of the gut microbiota of honey bees from sub-Saharan Africa and underscores the need to study microbiome diversity in other indigenous bee species and regions.

## 1. Introduction

Beekeeping has become an important tool to mitigate poverty, diversify income generation, and conserve biodiversity, in sub-Saharan Africa [1,2,3,4]. In Kenya, it provides critical pollination services, and contributes to food security and income generation for small-scale farmers in semi-urban and rural areas [5,6,7,8]. For instance, in Kakamega (densely populated farmland in Western Kenya), pollination services are estimated to generate about 40% of the annual crop market value [9]. Whilst there have been no reports of significant bee population decline in Kenya, common bee pathogens have been reported; thus, there is a need to assess the overall status of honey bee health in Africa [10,11].

Symbionts—beneficial associated microbes—have a profound effect on host physiology and fitness. In insects, these microbes play major roles in nutrition, protection against natural enemies, reproductive manipulation, communication, among others [12,13,14,15,16,17,18,19]. The honey bee, *Apis mellifera*, harbors a distinctive, simple, and specialized gut microbiota that play major roles in bee metabolism and health [20,21,22,23,24,25]. It is composed of nine bacterial phylotypes, five of which are found virtually in every honey bee worker [23,26,27,28,29]. Core members include the two Gram−negative bacteria *Gilliamella apicola* and *Snodgrassella alvi* that belong to the Gammaproteobacteria class and reside in the ileum, a zone between the midgut and the rectum [30,31]. *G. apicola* is located in the lumen and it is involved in sugar fermentation [22,25,31,32,33]. *S. alvi*, in contrast, is located at the vicinity of the gut epithelium and is a non-fermenting bacterium able to carry out aerobic oxidation of carboxylates, most likely, generated from *G. apicola* metabolism [22,30,31]. This complementary use of resources by different bacterial members, enables them to occupy the same niche, and it might be the result of a longstanding co-evolution [22,33,34,35].

Other highly abundant core gut microbiota include *Lactobacillus,* a vast genus that had its classification recently re-visited, and has been divided into 25 different genera [36]. In bees, members of the Firmicutes phylum group 4 and 5 have revealed to be the most widespread and abundant found in several species of corbiculate bees [27,29,32,37,38,39,40]. Some bee-isolated *Lactobacillus* strains have been shown to have antimicrobial properties against bee bacterial pathogens making them good candidates for use as probiotics [40,41,42,43]. *Lactobacillus* resides in the rectum together with *Bifidobacterium asteroids*, another core member that belongs to the Actinobacteria phylum [37,38,44]. Together, they have been shown to digest complex pollen aromatic compounds and ferment the derived sugars into organic acids [22,33].

The other four core gut microbiota members are present in almost every worker bee and have been shown to induce additional important phenotypes. *Frischella perrara* is a Gammaproteobacteria closely related to *G. apicola* that resides in a restricted part of the gut called the pylorus, a short region between the midgut and the ileum [45]. *F. perrara* activates a melanization cascade in the pylorus that results in a dark band on the epithelium surface, called “scab phenotype” [45,46]; however, the potential role of this phenotype remains elusive. Members of the Alphaproteobacteria that include Acetobacter species (*Commensalibacter* sp., Apha 2.1), *Bartonella apis* (Alpha 1), and *Bombella apis* (Alpha 2.2) are also part of the core honey bee gut microbiota. These symbionts are also present in several species of corbiculate bees and are involved in sugar metabolism [27,47,48]. Finally, other prevalent members are *Apibacter adventoris* and *Apibacter mensalis* from Bacteroidetes phylum isolated from honey bee and bumble bee intestine [49,50,51]. Together with *Lactobacillus* Firm-5 and *Gilliamella* genus, *Apibacter* abundance has been reported to correlate with lower loads of a trypanosomatid bee parasite in bumble bees, highlighting the potential role of this group in bee health [52].

While *A. mellifera* harbors a conserved and simple gut microbiota, recent findings have uncovered a large hidden diversity at the strain level [53,54]. Moreover, it has also been reported that honey bee gut microbiota is shaped by seasons (i.e., diapause), landscape, and host genetic background [55,56,57,58,59,60,61]. Bee workers that survive over the winter harbor in their guts higher loads and less diverse bacteria affected by their diet during this period [59]. Surprisingly, whereas beekeeping has been widely promoted as a tool to mitigate poverty in tropical and subtropical regions of the world, no comprehensive studies of honey bee gut microbiota have been done in these regions, where there are no pronounced seasons, and pollen/nectar resources are present all year long [1,6].

Most of the gut microbiota studies are based on the fully domesticated European honey bee that has endure human artificially rearing and breeding for centuries. Additionally, common artificial feeding practices in western countries have been reported to be detrimental for bee gut microbiota [62,63]. In contrast to the western situation, sub-Sahara African bee keeping remains rudimentary and modern techniques and infrastructure are still under relatively low levels of adoption. Moreover, it hosts a highly significant and poorly studied *A. mellifera* genetic diversity (e.g., in Kenya, only, there are more than five subspecies, [11,64]). The current sub-Sahara African indigenous and traditional management methods, involving very little human intervention and frequent bee exchange between semi-managed and feral colonies (trapping feral swarms and high rates of colony absconding)—might be associated with significant and uncharacterized gut microbiota diversity selected for by different evolutionary pressures [11,65].

Here we characterized the *Apis mellifera scutellata* gut microbiota from three different locations in Kenya. This is the first comprehensive study addressing the gut microbiota of honey bees from tropical sub-Saharan Africa and aims to establish the basis for future bee symbiont studies in this region. Profiling the diversity and effects of the local environment on the gut microbiota in Africa, constitutes an important resource to enhance bee health and climate change resilience; and thus sustainably promote beekeeping and bee services across the globe.

## 2. Materials and Methods

### 2.1. Honey Bee Sample Collection and Preparation

Honey bee (*Apis mellifera scutellata*) foragers were collected at the hive entrance in January 2018 from Western (Butali and Kalenda), Central (Githunguri, Kiambu, and Ruiru) and Coastal (Bahari, Matsangoni, Marenje, and Mwale) Kenya (Figure 1), from captive bred hives (i.e., feral bee swarms are collected by the beekeeper in Langstroth hives, absconding happens regularly). Ten forager worker bees per hive from 3 hives per site/apiary were collected and washed in 4% sodium hypochlorite, then 70% ethanol and finally 1× PBS for 2 min to eliminate any external microorganisms or contaminant DNA attached to the cuticle [66]. Entire guts were dissected aseptically using forceps and each gut placed in 2 mL microcentrifuge-tube containing 500 µL PBS. Samples were stored at −80 °C before DNA extraction. 

### 2.2. DNA Extraction

DNA extraction was done using the CTAB-Phenol–chloroform extraction method. Briefly, previously prepared intestinal homogenate tubes were placed in a 2 mL microcentrifuge-tube containing 750 µL CTAB solution (20 g CTAB in 1L CTAB base: 100 mL 1 M Tris-HCL pH 8, 280 mL 5 M NaCl, 40 mL 0.5 M EDTA pH 8, complete with MilliQ H_2_O to a liter), 2 µL b-Merc-ethanol and 100 µL–200 µL beads (3 mm diameter). Tissue homogenization was done at 30/s frequency for 3 min using the Tissue Lyser II QIAGEN and the homogenate was kept on ice. Then, 1 mL of phenol was added and mixed, and incubated at 64 °C for 6 min, then, placed on ice for 5 min and the supernatant transferred into a 2 mL microcentrifuge-tube containing 400 µL of chloroform and mixed by inverting several times, centrifuged at 13,000 rpm for 10 min at room temperature. The aqueous upper phase was transferred into a new microcentrifuge-tube. Then, 500 µL Phenol-Chloroform-Isoamyl alcohol 25:24:1 was added, mixed, and centrifuged at 13,000 rpm for 3 min at room temperature. The aqueous upper phase was transferred into a new microcentrifuge-tube and 500 µL of chloroform added again, then centrifuged at 13,000 rpm for 3 min at room temperature. The aqueous upper phase was transferred into a new microcentrifuge-tube and the DNA was precipitated using 900 μL of 100% ethanol and incubated overnight at −80 °C. Tubes were then kept on ice for 5 min to equilibrate temperature and centrifuged at 13,000 rpm for 30 min at 4 °C. The DNA pellet was washed twice with 1 mL of 70% ethanol and then dried at room temperature for 10 min. The DNA was re-suspended in 200 µL MilliQ water.

### 2.3. 16S rRNA Gene Amplification and Sequencing

Extracted DNA samples were quantified fluorometrically (Qubit, Thermo Fisher Scientific Inc., Waltham, MA, USA) and then diluted to 10 ng/mL for use as PCR template. Primers spanning the V4 region of bacterial 16S rRNA: 515f: 5′-GTGYCAGCMGCCGCGGTAA-3′; 806r: 3′-GGACTACNVGGGTWTCTAAT-5′ were used [67,68]. Triplicate of 25 µL reaction were amplified at an annealing temperature of 50 °C using Platinum ™ Taq DNA Polymerase High Fidelity enzyme for 25 cycles. Sequencing libraries were synthesized with the Kapa Hyper prep kit (Kapa Biosystems, Wilmington, MA, USA), according to the manufacturer’s instructions. PentAdaptersTM (PentaBase, APS, Odense, Denmark) were used to barcode the library and were diluted according to Kapa’s recommendation and the starting DNA concentration. After the final amplification step, libraries were then quantified using Qubit dsDNA BR Assay Kit (Thermo Fisher Sc., Waltham, USA, MA) and the fragment size was assessed on a Fragment Analyzer (Advanced Analytical Technologies, Inc. Ankeny, IA, USA). Finally, libraries were multiplexed and sequenced using paired end reads on a MiSeq instrument. All 16S rRNA gene amplification and sequencing was done at Center for Integrated Genomics, University of Lausanne, Switzerland (https://wp.unil.ch/gtf/, accessed on 01/05/2018). 

### 2.4. 16S rRNA Gut Community Analysis

QIIME 2 (version 2019.10) was used for the 16S rRNA sequences analysis. Briefly, demultiplexing was done followed by visualizing the quality of the reads and trimming primers using Cutadapt (version 2.10). Low quality read trimming, denosing, chimera filtering, and merging the paired end reads was done using DADA2 [69]. Taxonomic classification was performed against SILVA132 database using a pre-trained Naive Bayes classifier [70,71]. Data was rarefied and Alpha diversity was determined using Evenness, Faith_pd, Shannon and chao1 index, and the statistical differences of gut microbiota diversity across the locations were tested using Kruskal–Wallis H test. Average diversity values were plotted for each sample at each even sampling depth and samples were grouped based on sample metadata. In addition, PERMANOVA was performed to compare microbial communities between locations [72]. To visualize the clustering of the microbial communities per locations principal coordinates (PCoA) was done using Bray–Curtis distances (multivariate community analysis). To establish the relative abundance of gut microbiota at the genus level across the locations, an ANOVA test using the least significant difference (LSD) test was performed using the R statistical software (v 4.0.2, Vienna, Austria). Pearson correlation test was additionally performed to identify the impact of altitude and humidity on the gut microbiota abundances across the locations.

### 2.5. qPCR Analysis

*Gilliamella* absolute abundances were determined using quantitative PCR (qPCR) assays using specific primers targeting the 16S rRNA gene of *Gilliamella* and normalized to *actin*, the host housekeeping gene [22,66]. qPCR was carried out in on a Rotor-Gene Q ^®^ machine using 2 µL of 5× HOT FIREPol ^®^ EvaGreen ^®^ HRM Mix from Solis Biodyne, 0.5 µL of each primer, 5 µL of the DNA template and 2 µL of nuclease-free water. The amplification was conducted in the following steps: denaturation step at 95 °C for 15 min, followed by 40 cycles of denaturation at 95 °C for 15 s, annealing at 60 °C for the 20 s, and elongation at 72 °C for 20 s, followed by a final elongation step for 5 min at 72 °C. Samples were run in duplicates. Variations of *actin* amplification, as a proxy of the gut size and extraction efficiency was minimal (+/−0.5 Ct).

## 3. Results

### 3.1. Bacterial Community Members Associated with Kenyan Apis Mellifera Gut

We profiled the gut microbiota of 86 worker honey bee guts, from 9 apiaries from 3 different geographical zones in Kenya (Figure 1); Western Kenya (~1700 m a.s.l., 0°29’16.7” N 34°50’12.2” E and 0°27′23.4” N 34°46′34.8” E) characterized by fertile agricultural lands with small farm size in the proximity of a conserved native rainforest; Central Kenya (~1900 m a.s.l., 1°08’27.7” S 36°57’33.5” E, 0°59’49.4” S 36°43’11.6” E and 1°04’36.2” S 36°47’22.9” E characterized by high lands, intensive small farm agriculture and semi-urban lands; and Coastal Kenya (~0 m a.s.l., north: 3°23’18.3” S 39°55’30.8” E and 3°53’45.5” S 39°36’28.0” E, and south: 4°31’42.4” S 39°09’37.4” E and 4°33’20.4” S 39°07’42.7” E) characterized by small agricultural farms. We analyzed 2,117,668 reads (average per sample 24,698, min 6357, and max 76,320) from the V4 region of 16S rRNA and we identified 4748 amplicon sequence variants (ASVs) (Appendix A). We found the core honey bee gut microbiota members: genus *Gilliamella*, *Snodgrassella*, *Lactobacillus* (Firm-4 and Firm-5), *Bifidobacterium*, *Frischella*, *Commensalibacter*, *Bombella*, *Apibacter,* and *Bartonella* (Figure 2, [29,38,73]).

As reported in other continents, *Gilliamella* and *Snodgrassella* were the dominant members of the core gut bacteria in Kenya (30.4% and 21.4% of reads, respectively) and present in all worker bee sampled [25,55,59,74]. Interestingly, we found *Frischella* to be the third dominant genus with an average of 16.9% reads among locations in Kenya with a high prevalence, 94% (Figure 2). Additionally, we observed the scab phenotype induced by this bacterium in our samples (up to 34%, data not shown, [45,46]). The relative abundance of *Lactobacillus* Firm-5 and Firm-4 where lower than expected when compared to other studies at only 12.9% and 1.7%, respectively, and a prevalence of 100% and 85%, respectively [27,38,73,75,76]. Furthermore, we found *Commensalibacter* sp. (Alpha 2.1) at a prevalence of 81% with an abundance of 2.4%. We also found *Apibacter mensalis* with 3% of the reads on average, and a prevalence of 62%, which contrasts with other studies that described higher relative abundance in *Apis dorsata* and *Apis cerrana*, but not in *A. mellifera* [27,49,50,51].

Apart from the core gut microbiota members, our analysis uncovered minor taxa group that we grouped as “Others” in Figure 2 and Appendix A that collectively represent less than 10% of all reads. The relative abundance of each taxa was very low—the highest was *Pseudomonas* with 2.3%, the second highest *Acidovorax* with 2.1%, then *Spiroplasma* with 2%, and finally *Gluconobacter* with 1.7%. The four taxa together represent around 8% of the reads but their presence was very scattered and uneven among samples from the same hive and/or from the same apiary (see Appendix A with the list of 10 most prevalent taxa on the “Others” category).

Together, the specific relative abundances and prevalence of the core members in our samples highlight a potential specialized local community composition in tropical sub-Saharan Africa.

### 3.2. Variation of Honey Bee Gut Microbiota Across the Locations

Honey bee gut microbiota is relatively constant across populations and geographies worldwide [29,38,49,55,73,76]. However, some studies have revealed the influence of landscape exposure, forage type, and agrochemicals on honey bee gut microbial communities [58,61]. We analyzed the overall diversity of worker honey bee gut bacteria in the three study sites. We rarefied our data to 10,000 reads (Appendix A) and found that bacterial diversity, as measured by ASVs species richness and evenness, does not significantly differ among the sites (Figure 3). Whilst evenness of the samples is very similar (Evenness: *H* = 1.90, *p* = 0.39), richness is more marked in the coastal and western samples when phylogenetic distances are taken into account (Shannon: *H* = 0.81, *p* = 0.67 vs. Faith-PD: *H* = 5.49, *p* = 0.06, Figure 3). This finding is further supported by the higher numbers of ASVs of core members present in the coastal samples compared to the other two locations (Appendix A).

Additionally, PERMANOVA (beta diversity) results revealed that there was no significant difference in distribution of bacterial communities between the locations, (*R*^2^ = 0.03074, *p* = 0.198, Appendix A) and principal coordinate analysis also supported this conclusion (Figure 4).

The overall analysis showed there was no major variation of *A. m. scutellata* gut microbiota across different locations highlighting how conserved in honey bee gut microbiota even in very different ecosystems (high lands vs. coastal area). Nonetheless, we found that the coastal region presents higher relative abundance and diversity of core members compared to western and central lands. Additionally, we uncover less environmental/opportunistic bacteria in the bee gut (see Figure 2 “Others” and Appendix A), indicating that the coastal ecosystem is a more suitable environment for beneficial bacteria growth, and they might outcompete environmental/opportunistic bacteria.

### 3.3. Gilliamela Abundance is Affected by the Local Environment

We investigated variations in the gut microbiota at the genus level across all of our sampling locations (Appendix A). *Gilliamella*, the most dominant core member in our study, showed significance differences across the locations and was significantly dominant in the coast (relative abundance measured as number of reads/total reads, *p* = 0.05, Appendix A). We confirm this finding by performing an absolute quantification of *Gilliamella* using qPCR in all samples across locations (Figure 5A, *p* = 0.000027). *Snodgrassella*, *Lactobacillus* (Firm-4 and -5), *Bifidobacterium*, *Bartonella,* and *Bombella* had also higher number of reads in the coast samples although their relative abundance was not statistically significant (Appendix A). For a better understanding of the impact of environmental factors on gut microbiota abundances, we investigate potential correlations between *Gilliamella*—altitude and humidity. We found that *Gilliamella* abundance was negatively correlated with altitude (R = −0.87, *p* = 0.0024, Figure 5B) and positively correlated with humidity (R = 0.86, *p* = 0.0032, Figure 5C).

## 4. Discussion

We carried out an initial characterization of the *A. m. scutellata* gut microbiota in tropical regions and investigated the impact of different geographical zones on the diversity of bacterial communities and the relative abundance of bacterial members. Our results showed significant differences in relative abundance of core members in the tropics, compared to studies in other latitudes, and uncovered the importance of the local environment for bacterial community composition. Since the gut microbiota has a profound effect in bee health, this study lays a critical foundation for a better understanding of African honey bee fitness and resilience against local environmental stresses and therefore to secure the role of beekeeping as a sustainable tool to reduce poverty in sub-Saharan Africa.

Our results show the presence and high relative abundance of the core honey bee gut microbiota. This suggests that the core bacterial communities are mostly constant across latitudes. Nonetheless, we found that the Kenyan coastal region harbors a higher diversity and a higher relative abundance of core members than other sites in Kenya. In addition, we show that *Gilliamella* is significantly more abundant in the coastal region and that this abundance correlates (inversely) with altitude and humidity. Interestingly, our results show that the *Frischella* is prevalent and dominant in Kenya, while the genus *Lactobacillus* was proportionally less abundant than expected when compared to studies done in other latitudes [35,38,59,73].

We found that most of the core gut microbiota members are proportionally more abundant in the coastal region where they have also a greater number of amplicon sequence variants (ASVs). There is increasing evidence that temperature plays an important role in gut microbiota composition in invertebrates [77]. We noted that coastal apiaries were surrounded by small scale agriculture that was similar to central and western apiaries; the weather, in particular temperature and humidity are constantly higher throughout the year. We did not find any effect of temperature on the gut microbiota abundance (data not shown); nonetheless, a more exhaustive sampling across altitudes (which largely correspond to different temperatures) is needed to clearly rule out the effect on temperature in the overall bacterial community. We speculate that the coastal abiotic conditions might be more suitable for the bee core gut microbiota members, although we cannot exclude the effect of specific endemic flora that might have an impact on the bacterial community through pollen and nectar metabolism and nutrition. Unfortunately, the 16S amplicon sequencing resolution did not allow us to characterize this diversity at the genomic and metabolic levels; metagenomics analysis (e.g., shotgun sequencing) will be needed for a better understanding of the structure of these communities [53]. Our study highlights the importance of the ecological niche for bee gut bacterial composition and give the first general overview of bacterial residents of the sub-Saharan bee gut.

*Gilliamella* are the most dominant bacteria in Kenyan *A. m. scutellata* gut microbiota, which is analogous to what has been described in recent studies elsewhere [22,33,55]. *Gilliamella* is highly abundant at the coast where it shows the highest diversity of ASVs. The abundance of *Gilliamella* is negatively correlated with altitude and positively correlated with the humidity. It has been reported that seasonal variation influences *Gilliamella* abundance in Europe and North America [55,59,78]. Our study is the first to report such fluctuations among locations in tropics. It is noteworthy that *Gilliamella* abundance increases in honey bees that are chronically exposed to the insecticides Fipronil and Thiamethoxam, as well as to the herbicide glyphosate [79,80]. These chemicals are registered and in use in Kenya and likely also in the farms surrounding sampled apiaries. Further studies should address the effect of these chemicals on *Gilliamella* in Kenya and asses a potential role in chemical detoxification and/or chemical selection of this bacterium in the gut of the bees located in the proximity of the use of these chemicals in agriculture, which could enlighten possible links between *Gilliamella* abundance and bee health.

We found that *Frischella* has the third highest relative abundance and prevalence in Kenyan *A. m. scutellata* gut microbiota, in contrast to studies in other continents where its presence is highly scattered and low in abundance [27,58,59]. Additionally, we also found the “scab” phenotype caused by *F. perrara* inducing the activation of an immune response in the pylorus region of the gut [45,46]. A role for *F. perrara* in pathogen protection through immune priming induced by this bacterium against a local pathogen could be plausible; nonetheless, we did not find any correlation between *F. perrara* abundance/presence and a potential bacterial pathogen abundance/presence (e.g., *Serratia* or *Pseudomonas* (data not sown)*,* [23]). Future studies measuring infection of other pathogens in the same samples, such as fungi or trypanosomes, can help to elucidate any protective host phenotype of *F. perrara* in Kenya. *F. perrara* abundance has been also positively correlated with aged pollen consumption, impaired development, and increase mortality in honey bees in Arizona, USA [81]. Nonetheless, our samples were collected in highly productive hives and its prevalence was high among samples (94%), which does not suggest detrimental effects of *Frischella* in Kenya. Together, the significant abundance and the induced-phenotype indicate that *Frischella* is very likely to play an important role in Kenyan bee physiology.

The relative abundances of *Lactobacillus* (Firm-4 and -5) are quite low in our study compared to samples from the Europe, Australia, Asia, and USA [27,38,73,75,76]. Studies have shown significant reduction of *Lactobacillus* in response to fungicides and insecticides used in Kenya [80,82]. Our data does not allow us to determine if the relative low abundance of *Lactobacillus* Firm-4 and -5 in Kenya is due to complementary digestive roles performed by other bacteria (e.g., *Bifidobacterium* and Bacteroidetes), bee genetic background, absence of seasons, or chemical exposure.

Interestingly, *Apibacter* was prevalent in our samples. *Apibacter* have been described mostly in bumble bees and Asian honey bee species (*A. cerrana* and *A. dorsata*), but rarely in *A. mellifera* [27,49,51]. It is likely that *Apibacter* sp. may participate in simple sugar digestion and host gut parasite protection in *A. m. scutellata* [52]. Addressing its correlation with bee pathogen presence in the hive, and survival experiment with mono-inoculated lines with *Apibacter* sp., could bring further insight into their role in sub-Saharan African bees.

In this study, we provide an initial characterization of the gut microbiota of *A. m. scutellata* from sub-Saharan Africa. We showed that core members of the bee gut microbiota are highly conserved in the tropics compared to other latitudes, and we report specific and interesting diversities that are affected by the local environment. Our results highlight the importance of future studies of honey bee gut microbiota in Africa and in the tropics where native bee species are present, which will contribute to the understanding of the role of single members and the whole community in bee health. Additionally, traditional beekeeping practices that involve very little human intervention, which are still the most common practices in the continent, might have led to different selective pressures on the bee gut microbiota community. A better understanding of this indigenous bacterial biodiversity will not only continue to position the honey bee as an important model for gut microbiota research, but might also help address global challenges, such as bee decline and climate change resilient agriculture through enhancement of pollination services.

## Figures and Tables

**Figure 1 microorganisms-08-01721-f001:**
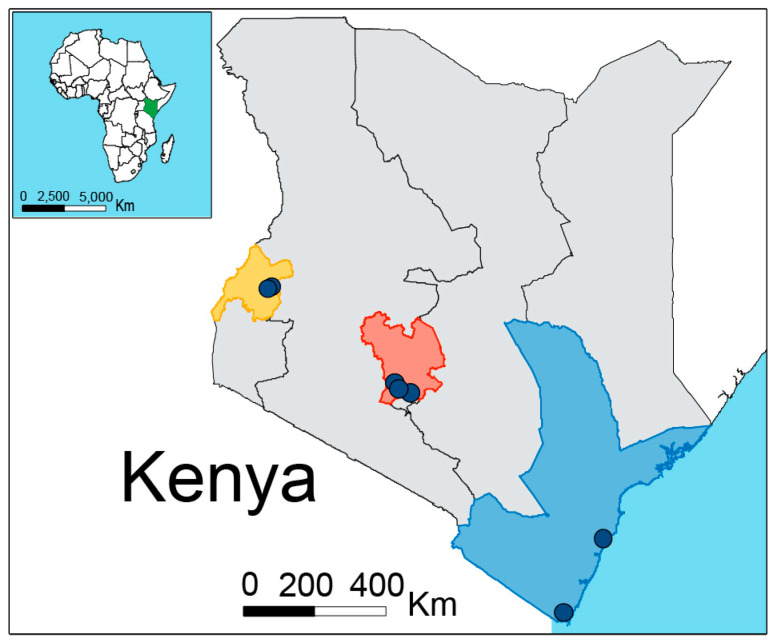
Sample collection map of *Apis mellifera scutellata* in Kenya. Western (yellow), Central (red) and Coastal (blue).

**Figure 2 microorganisms-08-01721-f002:**
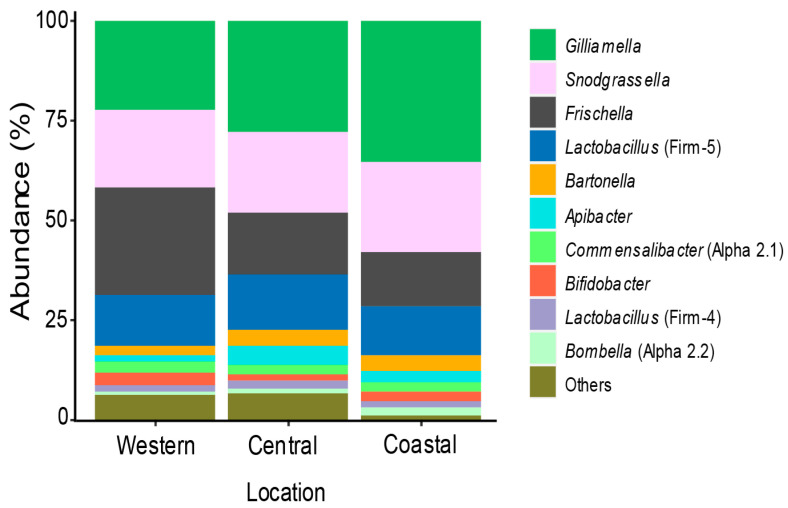
Bacterial genera associated with *Apis mellifera scutellata* gut in three different agro-ecological zones in Kenya.

**Figure 3 microorganisms-08-01721-f003:**
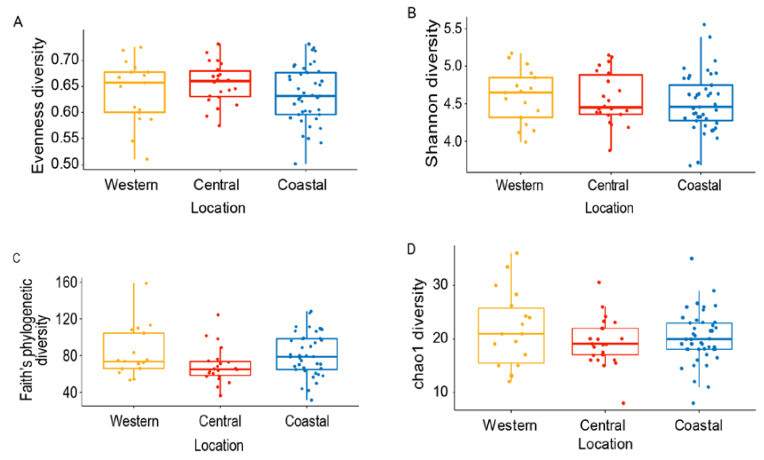
Alpha diversity analysis showed no bacterial community variation among locations. (**A**) Evenness diversity, (*p* = 0.39, *H* = 1.90), (**B**) Shannon diversity (*p*-value = 0.67, *H* value = 0.81), (**C**) Faith’s phylogenetic diversity, (*p* = 0.06, *H* = 5.49), (**D**) chao1 diversity (*p*-value = 0.54, *H* value = 1.2307) from Kruskal–Wallis H test.

**Figure 4 microorganisms-08-01721-f004:**
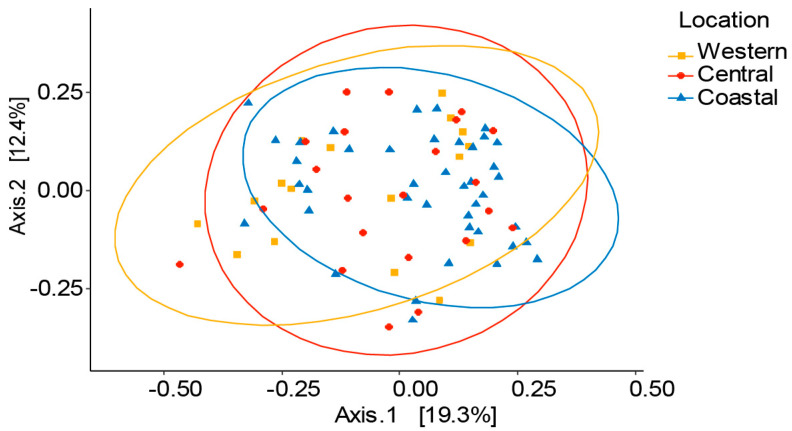
Beta diversity analysis showed no bacterial community variation among locations. Beta diversity (Principal coordinates) analysis based on Bray–Curtis distances (Multivariate community analysis). Different shades of each color indicate the distribution of the microbiota in respective to the locations.

**Figure 5 microorganisms-08-01721-f005:**
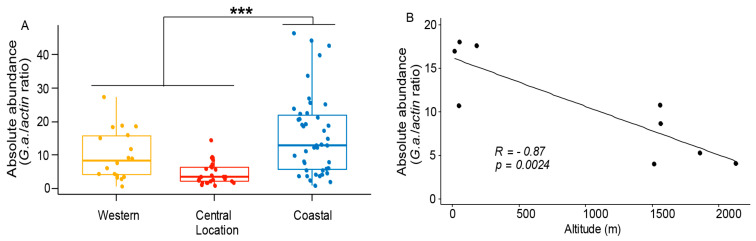
*Gilliamella* abundance varies across locations and significant correlates with the altitude and humidity. (**A**) *Gilliamella* absolute abundance across locations based on qPCR (*** *p* = 0.000027), (**B**) and (**C**) Correlation of *Gilliamella* abundance with the altitude and humidity per apiary, correlation coefficient (R) and *p* values were obtained using Pearson parametric correlation test.

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
