# Peer review of "Characterization of the Kenyan Honey Bee (Apis mellifera) Gut Microbiota: A First Look at Tropical and Sub-Saharan African Bee Associated Microbiomes"

_microorganisms, 2020, doi:10.3390/microorganisms8111721_

Round 1
Reviewer 1 Report
In General, the study was performed at a good level. Very little attention is paid in the manuscript to minor taxa of bacteria in the gut of bees. Despite the fact that there may be opportunistic microorganisms, the content of which may also differ between groups of honey bees. Authors should address this issue in more detail both in the results and in the discussion section.
There is also a lot of material transferred to supplementary materials. Therefore, it seems that there is not enough data in manuscript body. Think about what diagrams you can transfer to the manuscript from supplementary materials.
Author Response
Reviewer 1
Comments and Suggestions for Authors
In General, the study was performed at a good level. Very little attention is paid in the manuscript to minor taxa of bacteria in the gut of bees. Despite the fact that there may be opportunistic microorganisms, the content of which may also differ between groups of honey bees. Authors should address this issue in more detail both in the results and in the discussion section.
R: We have addressed reviewer’s concern, and we have discussed on the manuscript about the minor taxa as the reviewer suggested (L202-208).
There is also a lot of material transferred to supplementary materials. Therefore, it seems that there is not enough data in manuscript body. Think about what diagrams you can transfer to the manuscript from supplementary materials.
R: We have addressed the reviewer’s comment and have now included the alpha diversity (Figure 3) and Beta diversity (Figure 4) in the manuscript body.
Reviewer 2 Report
Hamba et al present a study in which the authors surveyed the gut microbiome of honey bees from multiple locations in Kenya using 16S rRNA sequencing. The topic of bee microbiome is interesting, especially as the authors note, from understudied regions of the world. While the study is largely descriptive, the authors present their findings well, and do not needlessly speculate on the results. Each section of the manuscript is understandable, although the discussion section is a little disjointed, and I encourage the authors to carefully check the readability of each paragraph. The study is well-designed and well-written, and in my opinion, should be published after revisions.
Specific comments (line numbers):
43- This is not true, as it is known that caterpillars lack a true microbiome, along with many solitary bees. See Hammer et al. 2019 (doi: 10.1093/femsle/fnz117). Please remove this.
44- It would be helpful to include some examples of this instead of only citations.
55- The genus Lactobacillus has been split into many new genera, and it would be worthwhile to include this information here as a note, as many new lactobacilli have new names. I do not suggest correcting the names throughout the manuscript. See Zheng et al 2020 (doi: 10.1099/ijsem.0.004107).
76-89- This paragraph is someone disjointed. There are two major points being made, one of the seasonality of the microbiome, and the other of beekeeping methods and ecology of honey bees. Could this be split up and have a little more information added to each? Each topic is interesting, but as it is doesn’t quite have enough detail to motivate the study.
77-79- Rothman, et al. 2018 (doi: 10.1007/s00248-018-1151-y) had a study investigating both seasonality and landscape resources. This should be cited here too.
92- This is a little oversold. Perhaps add “in this region” to the end of the sentence?
98- What was the source of the bees? Were they feral bees that were captured, commercially purchased, or captive bred?
98- Could you please provide the coordinates for the sites?
101- Is this 40% household bleach, or 40% sodium hypochlorite?
110-Typo: “QIAGEN”
139- Were reagent control or “blank” samples included as part of the run? Were contaminants filtered out, and if so, how? If not, how can the authors be sure their sequencing results are not affected by contamination?
163- Thank you for providing the coordinates!
170- Where the data normalized or rarefied in any way before running the diversity analyses? Generally alpha and beta diversity need equal sampling depth to be accurate. If not, could the authors please rerun the diversity analyses on rarefied/normalized data?
171- There are a fair amount of ASVs in the samples (over 11,000?), which might indicate adapters or primers being left in the sequences. Could the authors please double check and verify that they removed the adapters and primers?
175- The results are compositional and not absolute abundances, so saying “more” or “less” abundant is misleading.
203- Please add the R2 value (variation explained) for the PERMANOVA.
215 (Fig 3)- What are the units on the Y axis?
220- This is a little inaccurate, as unless qPCR was done, “abundance” cannot be measured. Are these results based on reads, or relative abundances? If reads, it would be useful to compare relative abundances instead.
223- As 16S rRNA data are composition, using an ANOVA on counts is not statistically accurate. For the differential abundance testing of genera across locations, could the authors please run analyses using compositional data tools such as Gneiss or ANCOM? These should both be run through QIIME2 or R as the authors used for other analyses.
245- This needs a citation.
247- If it was not significant, then you did not find a higher richness.
248- What does this mean?
272- This is a good point.
319- Thank you for including the complete ASV table.
Author Response
Reviewer 2
Comments and Suggestions for Authors
Hamba et al present a study in which the authors surveyed the gut microbiome of honey bees from multiple locations in Kenya using 16S rRNA sequencing. The topic of bee microbiome is interesting, especially as the authors note, from understudied regions of the world. While the study is largely descriptive, the authors present their findings well, and do not needlessly speculate on the results. Each section of the manuscript is understandable, although the discussion section is a little disjointed, and I encourage the authors to carefully check the readability of each paragraph. The study is well-designed and well-written, and in my opinion, should be published after revisions.
R: We thank the reviewer for his positive comments about our manuscript, his thorough review, and important additions/corrections he/she suggested. We believe we have addressed 100% or his/her comments. Please find below the answers for each point.
Specific comments (line numbers):
43- This is not true, as it is known that caterpillars lack a true microbiome, along with many solitary bees. See Hammer et al. 2019 (doi: 10.1093/femsle/fnz117). Please remove this.
R: We agree with reviewer’s comments and we have rephrased this sentence (L43-45).
44- It would be helpful to include some examples of this instead of only citations.
R: We have included examples of insect symbiont mediated phenotypes (L43-45).
55- The genus Lactobacillus has been split into many new genera, and it would be worthwhile to include this information here as a note, as many new lactobacilli have new names. I do not suggest correcting the names throughout the manuscript. See Zheng et al 2020 (doi: 10.1099/ijsem.0.004107).
R: We have included this information in the manuscript (L56-57).
76-89- This paragraph is someone disjointed. There are two major points being made, one of the seasonality of the microbiome, and the other of beekeeping methods and ecology of honey bees. Could this be split up and have a little more information added to each? Each topic is interesting, but as it is doesn’t quite have enough detail to motivate the study.
R: We have split this paragraph in two and added more information (L78-96).
77-79- Rothman, et al. 2018 (doi: 10.1007/s00248-018-1151-y) had a study investigating both seasonality and landscape resources. This should be cited here too.
R: We have added the suggested reference (L81).
92- This is a little oversold. Perhaps add “in this region” to the end of the sentence?
R: We have added “in this region” to this sentence (L99-100).
98- What was the source of the bees? Were they feral bees that were captured, commercially purchased, or captive bred?
R: We have added this information in the materials and methods section (L107-108).
98- Could you please provide the coordinates for the sites?
R: Coordinates are provided in the Results section (L180-186).
101- Is this 40% household bleach, or 40% sodium hypochlorite?
R: It is 4% sodium hypochlorite, we have corrected this sentence (L109).
110-Typo: “QIAGEN”
R: corrected (L119)
139- Were reagent control or “blank” samples included as part of the run? Were contaminants filtered out, and if so, how? If not, how can the authors be sure their sequencing results are not affected by contamination?
R: We included in the run a “blank” sample (in this tube, the biological material (dissected gut) was replaced by our working laboratory 1x PBS and followed all the extraction steps as a regular sample). We recovered very little bacterial species in this sample mostly Sphingomonas sp. with a total of 56 reads (>0.00002%). All this reads were filtered out in all samples before analysis.
163- Thank you for providing the coordinates!
R: You are welcome.
170- Where the data normalized or rarefied in any way before running the diversity analyses? Generally alpha and beta diversity need equal sampling depth to be accurate. If not, could the authors please rerun the diversity analyses on rarefied/normalized data?
R: Our data was rarefied to 10,000 reads from the beginning. We have now included this information on the main text (L150 and 218-219) and the rarefaction curves in Figure S1.
171- There are a fair amount of ASVs in the samples (over 11,000?), which might indicate adapters or primers being left in the sequences. Could the authors please double check and verify that they removed the adapters and primers?
R: We thank the reviewer for noticing this mistake; we confirmed that all our reads were correctly trimmed from the beginning. The mistake about the 11,000 ASV comes from a copy paste mistake for a previous analysis. We had 4748 ASV only, we have corrected in the manuscript (L187).
175- The results are compositional and not absolute abundances, so saying “more” or “less” abundant is misleading.
R: We have changed “the third most abundant” to “dominant” that reflects that the quantification is relative, and we have made the corresponding changes across the manuscript.
203- Please add the R2 value (variation explained) for the PERMANOVA.
R: We have run the test again in R and provided the R2 values (L274 and Table S3).
215 (Fig 3)- What are the units on the Y axis?
R: The units were normalized number of reads. Since we have now plotted absolute abundance (qPCR data), units are now the ratio between targeted and housekeeping gene (see M&M and new Figure 5).
220- This is a little inaccurate, as unless qPCR was done, “abundance” cannot be measured. Are these results based on reads, or relative abundances? If reads, it would be useful to compare relative abundances instead.
R: We thank the reviewer for highlighting this issue; we have now included relative abundance and qPCR data that allow us to quantify Gilliamella in our samples and across regions. We have also done the respective text modifications on the manuscript (L251-262 and new Figure 5).
223- As 16S rRNA data are composition, using an ANOVA on counts is not statistically accurate. For the differential abundance testing of genera across locations, could the authors please run analyses using compositional data tools such as Gneiss or ANCOM? These should both be run through QIIME2 or R as the authors used for other analyses.
R: We perform the ANCOM test on our all data set, as suggested by the Reviewer, from QIIME2, our W-statistic was zero, and no significant features were identified for the percentile abundances (see attached graph).
When we tried to unravel why the results were a bit unusual, from this tutorial (https://forum.qiime2.org/t/ancom-giving-strange-w-values/1002/2) our dataset related to low counts. Since we could not add pseudocounts to replace the zeros due to their documented inherent introduction of bias in the data, we decided to filter our data to 99.2% abundance. At that level, we ended up with 706 ASVs. We tried to compute ANCOM analysis again but this time in R by following the script documented in https://github.com/FrederickHuangLin/ANCOM. Since there is no package in R that does ANCOM analysis, by following the script, we lost the region ‘Central’ from the output, which seem to be a problem of the script since it is newly in implementation in R. Surprisingly, even after filtering to the most abundant ASVs, the W- statistic was still zero (see attached graph).
We then thought to pick only the core bacteria after grouping them together but we later realized that ANCOM does not work well with grouped data as documented in this source https://forum.qiime2.org/t/questions-regarding-ancom-input-and-output/2304/2.
Finally, as an alternative solution, instead of using ANOVA on read counts, we did it on relative abundance (as suggested by the reviewer on the previous point). We found again that only Gilliamella was statically different across locations (Table S3). To confirm further our results, we quantify Gilliamella abundance using qPCR and we found that Gilliamella is statically more abundant in the coastal region and correlation values (R) with altitude and humidity using qPCR quantification remained significant (see TableS4).
245- This needs a citation.
R: References have been added (L252).
247- If it was not significant, then you did not find a higher richness.
R: See answer below
248- What does this mean?
R: Sorry, there was a mistake in the two sentences, it has now been corrected (L253-254)
272- This is a good point.
R: Thank you.
319- Thank you for including the complete ASV table.
R: You are welcome.

Reviewer 3 Report
It is well recognized, that gut bacteria communities of insects, including Apis spp., have been found involved in food digestion, nutrient provisioning, and they generally can contribute to host health through immune system stimulation as well as conferring resistance against pathogens. The ‘healthy-phenotype’ of gut microbiome of Apis spp. has raised legitimate interest in recent years because of high honey bee mortality, and because 70% of the plants we eat are insect-pollinated.
In the reviewed manuscript (microorganisms-974258-peer-review-v1) in culture-independent studies based on 16S rRNA sequences of the gut microbiome of healthy adult honeybees obtained from three different eco-regions of Sub-Saharan Africa, the Authors confirmed existence of core microbiota in Apis mellifera. They also showed differences in abundance of Gilliamella in gut of honey bees depended on two factors: humidity and altitude.
However, phenomena in nature are most often very complex: many factors affect one effect and one factor affects many effects. The Authors are fully aware of this and in the Discussion section they critically analyze their study. Besides, used methods, data analysis, and manuscript preparation were performed without reservations, and conclusions formulated are fully adequate to the performed analysis of statistical significance. Therefore, I recommend the manuscript for publication.
I have only one very minor remark that have no bearing on the high quality of the research and analysis:
l.106 Please specify before ‘tubes’ e.g. previously prepared intestinal tubes; in such or similar form it will be clear to non-native speakers
Author Response
Reviewer 3
Comments and Suggestions for Authors
It is well recognized, that gut bacteria communities of insects, including Apis spp., have been found involved in food digestion, nutrient provisioning, and they generally can contribute to host health through immune system stimulation as well as conferring resistance against pathogens. The ‘healthy-phenotype’ of gut microbiome of Apis spp. has raised legitimate interest in recent years because of high honey bee mortality, and because 70% of the plants we eat are insect-pollinated.
In the reviewed manuscript (microorganisms-974258-peer-review-v1) in culture-independent studies based on 16S rRNA sequences of the gut microbiome of healthy adult honeybees obtained from three different eco-regions of Sub-Saharan Africa, the Authors confirmed existence of core microbiota in Apis mellifera. They also showed differences in abundance of Gilliamella in gut of honey bees depended on two factors: humidity and altitude.
However, phenomena in nature are most often very complex: many factors affect one effect and one factor affects many effects. The Authors are fully aware of this and in the Discussion section they critically analyze their study. Besides, used methods, data analysis, and manuscript preparation were performed without reservations, and conclusions formulated are fully adequate to the performed analysis of statistical significance. Therefore, I recommend the manuscript for publication.
I have only one very minor remark that have no bearing on the high quality of the research and analysis:
l.106 Please specify before ‘tubes’ e.g. previously prepared intestinal tubes; in such or similar form it will be clear to non-native speakers
R: We thank the reviewer for his positive comments about our manuscript. We have added the suggested information (L115).
Round 2
Reviewer 2 Report
The authors have done a tremendous job of responding to my comments. I have no further comments or concerns, and I look forward to seeing this published.